# The macroeconomic effects of adapting to high-end sea-level rise via protection and migration

Gabriel Bachner [1] ✉, Daniel Lincke [2] & Jochen Hinkel [2,3]

Climate change-induced sea level rise (SLR) is projected to be substantial, triggering human adaptation responses, including increasing protection and out-migration from coastlines. Yet, in macroeconomic assessments of SLR the latter option has been given little attention. We fill this gap by providing a global analysis of the macroeconomic effects of adaptation to SLR, including coastal migration, focusing on the higher end of SLR projections until 2050. We find that when adapting simultaneously via protection and coastal migration, macroeconomic costs can be lower than with protection alone. For some developing regions coastal migration is even less costly (in GDP) than protection. Additionally, we find that future macroeconomic costs are dominated by accumulated macroeconomic effects over time, rather than by future direct damages, implying the need for immediate adaptation. Finally, we demonstrate the importance of including autonomous adaptation in the reference scenario of economic assessment studies to avoid overestimation of adaptation benefits.

Climate change is already visible via bio-physical impacts around the world, which are expected to increase, depending on future greenhouse gas emissions[1,2]. This will trigger a range of socio-economic consequences and risks. To adapt effectively, and as a motivation for climate change mitigation, it is key that policy makers understand these potential consequences as well as the socio-economic effects of adaptation.

The literature has identified sea-level rise (SLR) as one of the major risks from climate change[2,3]. Even when meeting the target of the Paris agreement of staying well below 2 °C of global warming, mean and extreme sea levels are projected to rise substantially during the 21st century[1]. It has also been shown that SLR will continue for further centuries and can only be slowed down but not avoided[4]. The regions facing highest risk from SLR are small island states, delta regions, and often regions of the global south[5–7], but SLR also poses significant risks for developed regions[8–10].

There are various adaptation options to combat SLR, which can be categorized into: advance (creating new land seawards), protection (blocking inland propagation of mean and extreme sea levels), retreat (giving up land and out-migration of people), accommodation (reducing vulnerability to flooding through, e.g., floodproofing buildings or early warning systems) and ecosystem-based solutions (supporting advance, protection and/or retreat through restoring and maintaining coastal ecosystems) (cf. Oppenheimer et al.[7]). It has been shown that adaptation in the form of coastal protection can be very effective at reducing impacts[11–13], and also that protection and advance is cost-efficient for densely populated and urbanized areas, but inefficient and very expensive relative to local GDP for rural and less densely populated areas[11,14–16]. Accommodation can be particularly effective for small rises in sea levels, but ceases to be so for higher SLR[7]. As a result, high SLR may trigger coastal retreat and the associated out-migration of people, a possibility that has received increasing attention in the literature. Accelerated SLR and increased coastal flooding has the potential to displace millions of people[17–21]. As an example, even under cost-benefit optimal protection decisions, it is estimated that 17 to 72 million people globally will have to migrate from coastal areas during 21st century[22].

[1]Wegener Center for Climate and Global Change, University of Graz, Graz, Austria. [2]Global Climate Forum e.V., Adaptation and Social Learning, Berlin, Germany. [3]Resource Economics Group, Thaer-Institute of Agricultural and Horticultural Sciences, Humboldt Universität zu Berlin, Berlin, Germany. ✉e-mail: gabriel.bachner@uni-graz.at

Given the potential major impacts of SLR, decision makers need to understand its macroeconomic effects, i.e., how SLR and different adaption responses affect whole societies and economies. On the one hand, this need has been addressed through fully integrated models (often called integrated assessment models, IAMs) which include simple representations of the economy (e.g., a simple growth model), the climate (e.g., the MAGICC[23] model) and impact systems (i.e., damage functions) in order to solve the climate and the economic systems' equations simultaneously (e.g., the DICE[24,25] or the FUND[13] model). The downside of this approach is the lack of detail and in particular the non-consideration or only very stylized consideration of adaptation, which constitutes a major limitation, as adaptation is the dimension to which SLR impacts are by far the most sensitive[26]. On the other hand, the need for macroeconomic assessments is addressed by connecting climate, impact and economic models in a soft-linked manner, i.e., a sequential passing-on of information from climate models to detailed sectoral models and typically to macroeconomic models at the end of the modelling chain (e.g., a computable general equilibrium (CGE) model as for example in Parrado et al.[27]). As opposed to fully-integrated IAMs, which are rather coarse and highly aggregated[28], the soft-link approach allows for a more detailed impact- and sector-wise assessment, enabling us to look into the distribution of effects of impacts and different types of adaptation across regions, sectors and even households[29]. In addition, it allows for differentiation between direct and indirect effects via market interconnections and thus for drawing conclusions as to how much localized climate shocks are amplified or absorbed within the economic system (see e.g., Hallegatte et al.[30]).

There are, however, two major limitations in the macroeconomic literature that we address in this paper: First, the existing literature typically assumes a no-adaptation reference scenario[27,31–34], which means that coastal societies neither raise coastal protection nor retreat from the floodplain as sea levels rise. Clearly, this is an unrealistic assumption, because people have been upgrading coastal protection as response to local SLR and other coastal hazards for centuries in the past and if this fails, people will not just simply stay in the floodplain experiencing higher and higher floods every year[7,35,36]. Second, the adaptation scenarios considered in the macroeconomic literature so far predominantly focus on coastal protection, thus disregarding retreat, the other major adaptation response to be expected, as argued above. We acknowledge that a few macroeconomic studies do include migration; some of the IAM literature based on the FUND model has considered retreat[12,37], but using a stylized national-level damage function in which extremes are only included implicitly, which according to the empirical evidence are the main drivers of retreat[20]. Further, FUND solves for an optimal outcome under perfect-foresight (resembling managed retreat by a perfectly informed social planner) and thus also captures migration in a highly stylized way and not in the form of reactive migration. Yet these studies indicate that the costs of SLR-induced displacement are substantial, even in optimal outcomes. Further, Pycroft et al.[33] use a CGE model and include migration in their analysis. However, migration is modelled as forced consumption, thus ignoring productivity losses from losing and moving capital, and—as they use a static model—endogenous dynamics over time are not accounted for (see Tol et al.[38] for a critique). A key finding of Pycroft et al. is, that the derived welfare losses from SLR increase substantially when using a broader set of impacts, e.g., by also including migration costs. Uniquely, the work of Desmet et al.[39] considers migration, but without considering protection, which results in unrealistically high numbers of migrants. It also does not include sea flood cost. One interesting aspect of the study is that it considers the additional losses in economic performance due to the dispersion of spatial economic agglomerations, which is beyond the scope of models that are not spatially explicit, such as CGE models and IAMs.

In the analysis presented here we address both of these limitations and contribute to the soft-linked assessment literature. Specifically, we include—alongside a no-adaptation scenario—combinations of protection and retreat as adaptation options, allowing us to test the importance of including autonomous retreat in the reference scenario (instead of assuming no adaptation). We use a model compound that connects the detailed bottom-up coastal impact and adaptation model DIVA with the multi-sectoral and multi-regional global CGE model COIN-INT. Protection is typically a form of anticipatory, publicly planned, and capital-intensive adaptation, and hence we model it as such. Conversely, out-migration is interpreted as an ad-hoc individual retreat and thus is an example of reactive, private and autonomous adaptation[40,41], with costs arising due to moving and rebuilding capital stocks. We regard these two possibilities as contrasting cases and thus select them as key adaptation assumptions in our scenarios.

Modelling migration is notoriously complex, because it depends on a myriad of social, economic, environmental and political push and pull factors, as well as their complex interplay[42–44], which cannot be captured in a coastal impact model. To avoid this complexity, we focus on modelling out-migration due to SLR only, following Lincke and Hinkel[22]. In terms of the migration destination, we follow general findings of the literature in that migration is mostly country-internal and that internal migration will rise[44,45]. In our autonomous adaptation scenario, we thus assume that SLR does not induce international migration, but that people and assets relocate within their country borders to locations that are not exposed to (current and future) SLR. The resulting costs are those associated with moving and rebuilding capital stock. Potential changes in economy-wide performance due to changes in spatial structures and agglomeration (as analyzed by Desmet et al.[39]) are assumed to be neutral.

We embed our analysis in the RCP-SSP framework[46,47]. Specifically, the three following scenarios are analyzed until 2050, with stated SLR as compared to 2015: First, RCP8.5-SSP5 with a high-end ice melting sensitivity assumption, leading to 0.39 m SLR by 2050 (1.62 m by 2100). Note, that by high-end we mean the higher end of projected scenarios. Second, RCP8.5-SSP5 with medium ice melting, leading to 0.19 m SLR by 2050 (0.63 m by 2100). Third, to put the former two scenarios into perspective, RCP4.5-SSP2, which is regarded as a middle-of-the-road scenario leading to 0.16 m SLR by 2050 (0.45 m by 2100). Comparing the changes in sea levels clearly shows that already in 2050, the first scenario is in fact a high-end scenario, as SLR is higher by a factor of 2, compared to the medium ice melting scenario of the same RCP-SSP combination. All three scenarios are run for the four different adaptation cases of (i) no adaptation, (ii) planned adaptation-only (via sea dikes), (iii) autonomous adaptation-only (via migration), and (iv) planned and autonomous adaptation combined (i.e., sea dikes and migration). All adaptation cases are compared to an SSP-specific baseline scenario that simulates the socio-economic development without any climate change (see Methods for detailed specifications). By comparing the different scenarios to the baseline, we can isolate the effect of climate change-induced SLR under different assumptions of adaptation. By comparing the different adaptation cases to each other, we can learn about the macroeconomic effectiveness of adaptation.

In this work, we find that adaptation clearly pays off from a macroeconomic perspective, but also that even with adaptation residual damages in terms of GDP losses might be substantial. Furthermore, under high-end SLR the combination of planned protection (sea dikes) and autonomous retreat (migration) is more cost-effective than relying solely on protection for regions such as India and South-East Asia. We explain this lower macroeconomic effectiveness of the protection-only strategy by the high necessary investments for sea dikes, which binds capital that would be used more productively elsewhere in the economy if protective adaptation requirements were lower. We also conclude that autonomous adaptation in the form of reactive coastal migration alone already reduces macroeconomic

losses substantially compared to a hypothetical case without any further adaptation, especially in developing regions. This implies that when autonomous adaptation is taken as a reference scenario, instead of a hypothetical and implausible no-adaptation scenario as is often the case in the literature, the macroeconomic benefits of protection (dikes) are much smaller. This implication is important for cost-benefits analysis, as the benefits of planned protection might be overestimated. We conclude that since some form of autonomous adaptation will happen in any case, economic impact assessments of SLR should use an autonomous adaptation scenario as a reference scenario, for example in terms of autonomous retreat from coastlines as presented here. The data needed for this is now openly available and should be taken up broadly to increase reliability of results. Lastly, we conclude that future economy-wide damages (measured in GDP loss) will be a multiple of future direct damages, since capital stock dynamics lead to a propagation of damages over time and to persistence of impacts[48]. Thus, indirect and intertemporal effects dominate the future macroeconomic costs of SLR, implying that if damages can be avoided now, future macroeconomic losses of climate change can be lowered substantially.

## Results

### Direct costs
Increasing average annual direct damages due to rising extreme sea levels (sea flood costs hereafter) are expected to be much lower in European regions than in Rest of the World (ROW) regions (see Supplementary Figs. 15–20). China (CHN) especially would experience very high damages, if no (further) adaptation were to be put in place. In Europe, regions with long coastlines are especially vulnerable, i.e., Italy (ITA), Northern Europe (NEU), United Kingdom (UKD) as well as the Mediterranean and South-eastern Europe (MEU). We see that both planned adaptation (protection), as well as autonomous adaptation (migration), are very effective in preventing severe increases in sea flood costs. With autonomous adaptation, costs first increase and start to flatten (or even decline) as of 2035, showing its reactive nature. When looking at the difference between the no adaptation and autonomous adaptation-only scenario, we see a large difference, especially for the ROW regions. By design, migration costs are zero in the cases without autonomous adaptation (without migration, see Supplementary Figs. 21–26). In the two cases where autonomous adaptation is involved, we see marked increases and peaks in migration costs from 2030 onwards. In the autonomous adaptation-only case, migration costs are especially high for the European regions of Italy (ITA) and Mediterranean and South-Eastern Europe (MEU), whereas in the ROW it is the emerging economies of Asia (ECA), China (CHN), North America (NAM) and the Oil Exporting Regions (OIE) that would experience the highest migration costs.

### Macroeconomic effects
Macroeconomic losses as a result of direct costs in the hypothetical case of no further adaptation are very high, due to sea floods destroying large amounts of essential means for production, such as buildings and machinery. Under the high-end scenario (i.e., RCP8.5-SSP5 high-end ice melting), GDP in the European regions of Italy (ITA) and Northern Europe (NEU) for example, would be 4.5% lower than in the baseline in 2050. On the other hand, for landlocked regions such as Central Europe (CEU) slight positive effects due to comparative advantages on global markets emerge. In the ROW regions the negative effects are even stronger, with GDP losses in 2050 of up to 11% in the Emerging Economies of Asia (ECA), followed by China (CHN), Oil Exporting Regions (OIE), Australia and New Zealand (AUZ) and South-East Asia (SEA) with GDP losses ranging between 9 and 7% (see Figs. 1a, 2a, and 3).

When continuing planned adaptation by raising sea dikes, relative GDP losses in 2050 (compared to baseline levels) remain below 1% in

European regions and below 2.5% in ROW regions. When assuming autonomous adaptation in the form of coastal migration only, relative GDP losses are limited to 3% in all regions, with many regions experiencing losses of less than 1%. In the combined case of autonomous and planned adaptation the results for European regions are not very different from the planned adaptation-only scenario, since protection measures (sea dikes) in this case protect mostly densely inhabited areas and thus avoid the need for migration. Interestingly, for all ROW regions the combination of planned and autonomous adaptation is at least as or even more effective than relying on planned adaptation (protection) only. In all ROW regions GDP losses in 2050 would not go beyond 1%.

The effects of the combined adaptation scenario show that in Europe, more populated areas are protected by dikes than in other regions and that in ROW regions relocated assets don't have to be protected by large unproductive investments (i.e., dikes) in the future. The higher effectiveness of the combined scenario is particularly visible in South-East Asia (SEA) and Oil-Exporting Regions (OIE). What also becomes clear is that autonomous adaptation in the form of coastal migration is more of a reactive process to rising sea levels. In Figs. 1a and 2a in the first years of the time horizon until 2035, the shape of the curves is very similar to the no adaptation scenario; only subsequently will sea levels rise enough to trigger migration. This becomes visible as a flattening or stabilization of the curves, or in some cases even a reversal of the trend.

### Amplification ratios
A common metric to illustrate the relative size by which direct damages are absorbed or amplified by propagating through the economic system within a year and by capital accumulation dynamics over time is the amplification ratio (AR, see Methods). An AR > 1 means that annual GDP losses are higher than direct costs from SLR in the same year; the value of the AR indicates the magnitude of this relationship. We show the development of ARs over time in Figs. 1b and 2b. In the first years of the analysis, direct damages can be partly absorbed by flexibility in trade, production processes and consumption behaviour, indicated by ARs < 1. However, as time passes, ARs become greater than 1 and reach relatively high levels in some cases (up to 17 for the Netherlands (NDL) in 2050). This trend is due to the dynamics of weakened capital accumulation over time, as SLR reduces the capital stock itself as well as capital income which slows down investment activities. Put differently, a loss of GDP (income) in one time step leads to lower investment and thus to a lower capital stock (and income) in the next time step. Direct damages are thus accumulating in terms of indirect damages over time. In addition, protective adaptation investment does not add to the productive capital stock of the economy, thus also lowering future income in terms of capital rents. Note that some regions show negative ARs, indicating that GDP losses are in fact negative (i.e., GDP gains), despite potential positive direct costs. In the migration scenarios we see different shapes and occasionally very high ARs, as while direct losses become relatively small due to previous relocation of assets, it coincides with lower GDP due to the described weakened capital accumulation effect. Macroeconomic costs thus persist throughout time.

### Sensitivity of impacts
As the largest uncertainty related to SLR is associated with how much the melting of the ice sheets of Greenland and Antarctica contribute to SLR, we first look at how sensitive macroeconomic effects are to such ice melting assumptions. We find that for RCP8.5-SSP5 with medium ice melting, relative GDP losses without adaptation would be much lower (up to 2% in European and up to 4% in ROW regions) and when assuming adaptation, losses can be reduced to similar levels as under high-end SLR (see Supplementary Figs. 27 and 28). Figures 4 and 5 show relative GDP effects as well as ARs for the RCP4.5-SSP2 medium

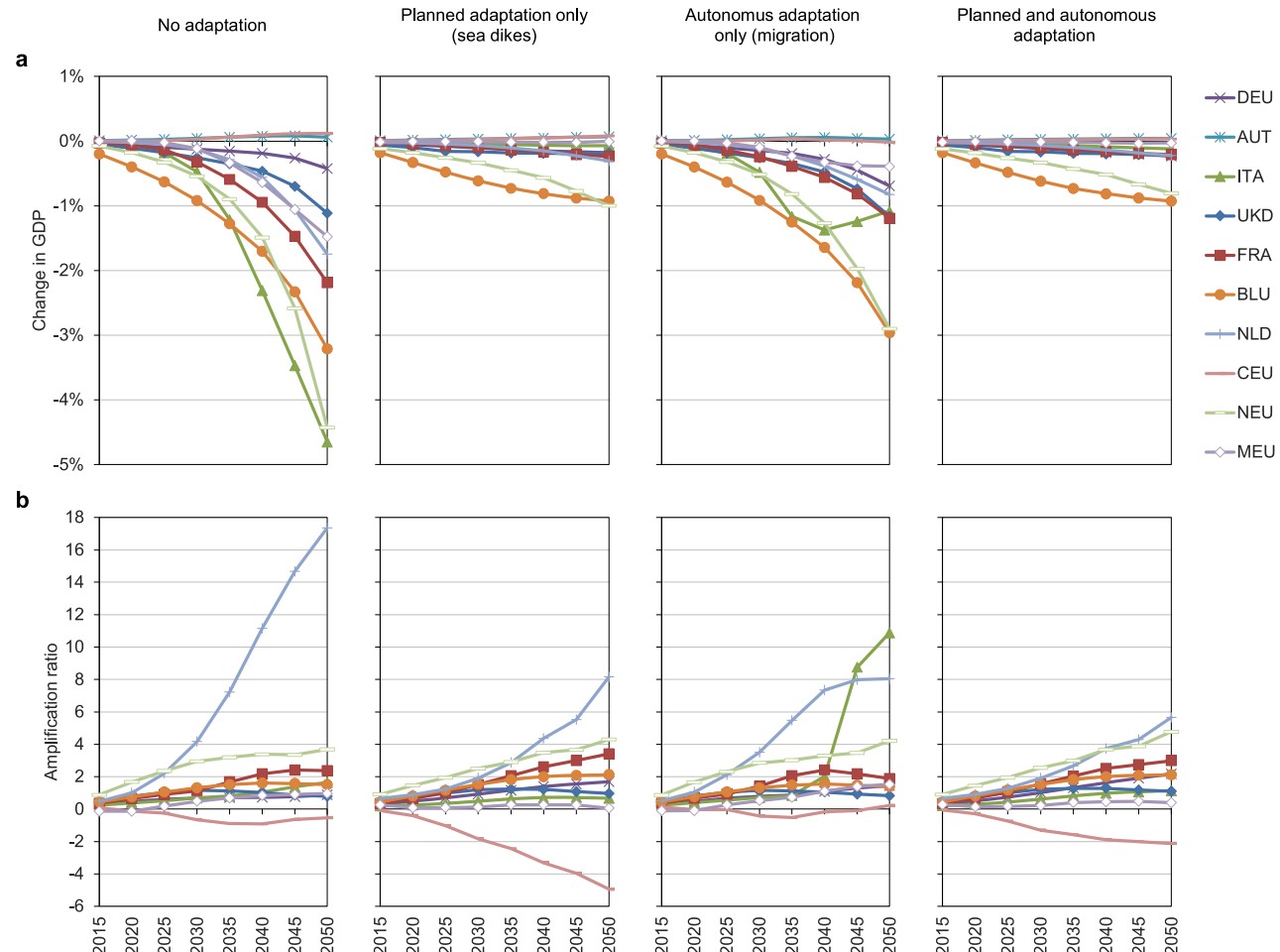

**Fig. 1 | GDP effects and amplification ratios for European regions.** Results under RCP8.5-SSP5 high-end sea-level rise, relative to baseline scenario, for four cases of adaptation. **a** Change in GDP. **b** Amplification ratios. Region abbreviations: DEU Germany; AUT Austria; ITA Italy; UKD United Kingdom; FRA France; BLU Belgium and Luxemburg; NLD Netherlands; CEU Central EU 27 + Switzerland; NEU Northern EU 27+ Liechtenstein, Norway and Iceland; MEU Mediterranean and South-eastern EU 27.

ice melting scenario. Interestingly, results are very similar to the RCP8.5-SSP5 medium ice melting scenario, indicating that ice-melting sensitivity is also the dominating uncertainty at a macroeconomic level until 2050 (see Supplementary Fig. 29 for a direct comparison of all three scenarios for 2050). Looking at the AR, we observe similar magnitudes, indicating that amplification and persistence of effects over time is also very large in moderate scenarios. Finally, we test the sensitivity of migration costs by decreasing (increasing) it by a factor of 0.5 (2). We find that results are robust and that sea flood costs are also the dominating component in the autonomous adaptation cases (see Supplementary Figs. 31–33).

**Comparison of protection and migration**

We now investigate in greater depth the difference between planned protection (sea dikes) and autonomous retreat (migration). Specifically, we compare their macroeconomic effectiveness in terms of avoided GDP loss relative to the baseline in 2050 by calculating the differences of relative GDP losses between two adaptation cases: planned adaptation-only and autonomous adaptation-only. Figure 6 shows these results for high-end and medium SLR under RCP8.5-SSP5 (see Supplementary Fig. 30 for RCP4.5-SSP2). We see that the bars are mostly positive, ranging between 0 and 2.5%-points, which means that GDP losses with planned adaptation are less severe than with autonomous adaptation. Interestingly, under high-end ice melting this is not the case for India (IND) and South-East Asia (SEA), where

we observe negative values, i.e., less severe losses under autonomous adaptation-only compared to planned adaptation-only. This can be explained by the underlying high investment requirements for planned protection measures, which do not add to the economy-wide productive capital stock. Put differently, sea dikes do not contribute to economic activity as a production factor that earns an income, as opposed to other capital such as machinery or buildings. The only benefit from this type of adaptation capital in the form of dikes is the reduced damage from flooding and this otherwise unproductive adaptation capital stock is much higher when following the planned adaptation-only strategy.

Ultimately, our scenarios allow for determination of the degree of change in the benefits of adaptation when the point of reference is changed to an autonomous adaptation scenario (instead of a hypothetical no-adaptation behaviour). For that, we perform two comparisons. First, the comparison of planned adaptation-only versus no adaptation, as is common in the literature. Second, the comparison of the combined case of planned and autonomous adaptation versus autonomous adaptation-only. For each comparison we calculate the difference in relative GDP loss as percentage point differences, i.e., the benefit of adaptation in terms of avoided relative GDP loss, however with different assumptions regarding the inclusion of autonomous adaptation. Each bar in Fig. 7 shows this benefit, indicating that the benefits of adaptation are much lower when including autonomous adaptation in the form of migration in the reference point (i.e., the light

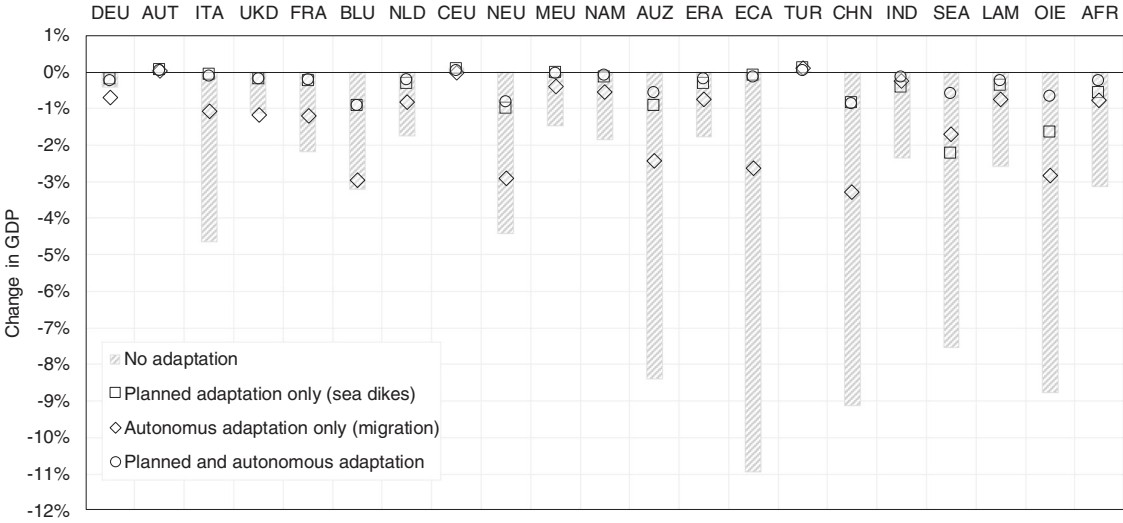

**Fig. 2 | GDP effects and amplification ratios for Rest of the World.** Results under RCP8.5-SSP5 high-end sea-level rise, relative to baseline scenario, for four cases of adaptation. **a** Change in GDP. **b** Amplification ratios. Region abbreviations: NAM North America; AUZ Australia and New Zealand; ERA Eurasian countries; ECA Emerging economies- Asia; TUR Turkey; CHN China; IND India; SEA South-East Asia; LAM Latin America (w/o Venezuela); OIE Oil exporting countries (OPEC: Middle East and Africa + Venezuela); AFR Africa. * = excluding regions AFR, LAM and OIE with ARs of −16,000, 81, and 40 in 2050 respectively.

**Fig. 3 | Relative GDP losses in 2050 under different adaptation scenarios.** Results under RCP8.5-SSP5 high-end sea-level rise, relative to baseline scenario, for four cases of adaptation. Region abbreviations: DEU Germany; AUT Austria; ITA Italy; UKD United Kingdom; FRA France; BLU Belgium and Luxemburg; NLD Netherlands; CEU Central EU 27 + Switzerland; NEU Northern EU 27+ Liechtenstein, Norway and Iceland; MEU Mediterranean and South-eastern EU 27; NAM North America; AUZ Australia and New Zealand; ERA Eurasian countries; ECA Emerging economies- Asia; TUR Turkey; CHN China; IND India; SEA South-East Asia; LAM Latin America (w/o Venezuela); OIE Oil exporting countries (OPEC: Middle East and Africa + Venezuela); AFR Africa.

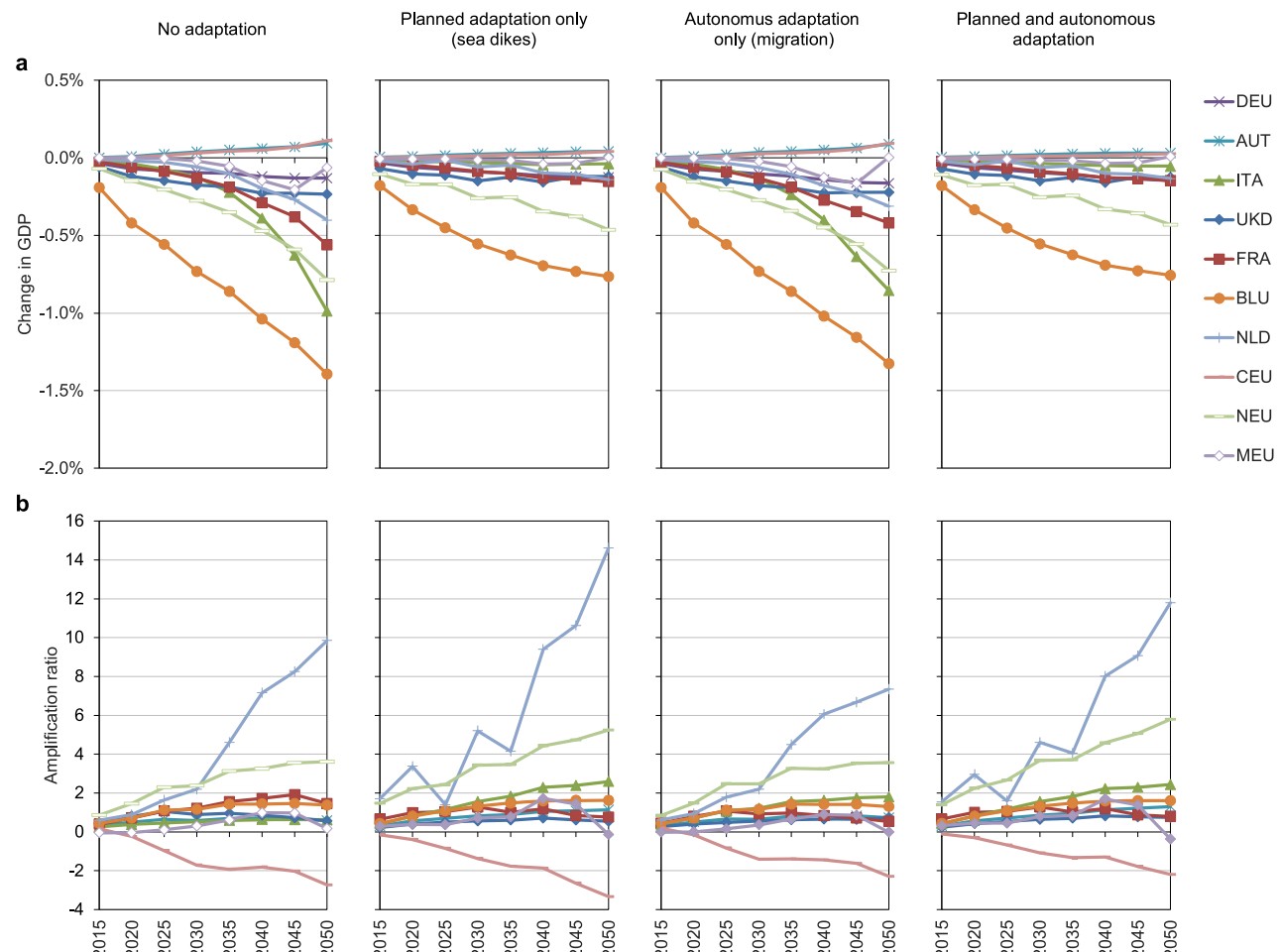

**Fig. 4 | GDP effects and amplification ratios for European regions.** Results under RCP4.5-SSP2 medium sea level-rise, relative to baseline scenario, for four cases of adaptation. **a** Change in GDP. **b** Amplification ratios. Region abbreviations: DEU Germany; AUT Austria; ITA Italy; UKD United Kingdom; FRA France; BLU Belgium and Luxemburg; NLD Netherlands; CEU Central EU 27 + Switzerland; NEU Northern EU 27+ Liechtenstein, Norway and Iceland; MEU Mediterranean and South-eastern EU 27.

bars are much shorter than the dark bars). These findings still hold when assuming only medium SLR, however, are not as pronounced.

## Discussion

Putting our results into perspective by comparing them to other recent studies on SLR using the same soft-link CGE approach, we find similar quantitative effects for those scenarios where a direct comparison is possible in terms of scenario choice and recentness of data[27,34]. However, it bears noting that the diverse studies on the macroeconomic consequences of SLR have considered different direct costs and mechanisms as to how these costs propagate through the economy, making direct comparisons difficult.

One limitation apparent in our study is that we measure macroeconomic effects on an aggregate level, i.e., by using GDP as an indicator and do not consider the distribution of cost burdens among different actors, most notably between the public and the private sector. Planned adaptation is often a public activity (e.g., building sea dikes, land use planning, building standards and codes, flood warning systems, and emergency planning), which needs to be financed out of scarce public resources or debt, and often a trade-off between effectiveness and long-term running costs needs to be made. Publicly financed adaptation could reduce the provision of public goods and services (e.g., health care, education etc.) thereby leading to unwanted distributional effects. Conversely, autonomous adaptation is often done in the private sector and thus costs are private, however, adaptation capacities might not be large enough to enable raising needed funds for private adaptation. Our results thus provide first insights as to an overall regional economic level; however, they need to be interpreted with care as GDP losses might also be—in addition to the presented level effects—structurally different (e.g., different effects on private and government consumption as well as different societal groups).

The assumption of migration as the only form of autonomous adaptation in our study seems to be simplistic, however, it allows for a global assessment and enabled us to reveal a substantial bias in results when estimating the benefits of protective adaptation, especially for high-end SLR. We argue that the assumption of out-migration as a form of autonomous adaptation (out of many) is more plausible than the usually applied no-adaptation assumption. As there is very little empirical evidence of the link between migration and SLR to date, autonomous migration is very difficult to parameterize. Our choice of a high flood probability threshold (1-in-1 year flood return level) as threshold for autonomous migration can be seen as a model for the most reactive form of autonomous adaption and can thus be seen as a limit case. The sensitivity of the migration numbers to changes in this threshold (as well as to sensitivity is to changes in the assumption on migration unit cost) has been analyzed by Lincke and Hinkel[22]. A limitation of this study is that the assumptions about what places will be protected (based asset density) are not tested for sensitivity.

Another limitation of our study is that we did not consider other direct costs incurred by the impacts of SLR such as costs due to

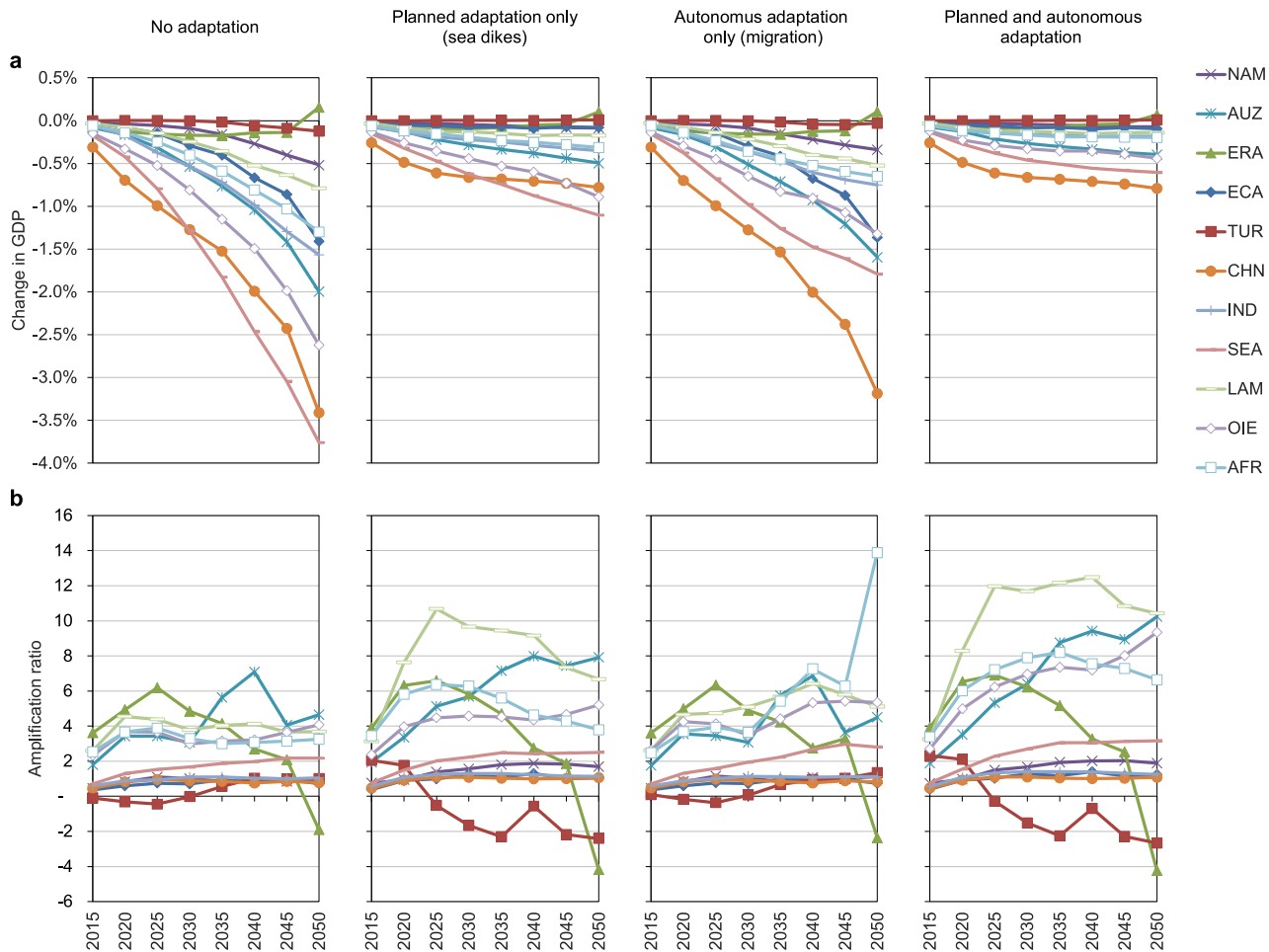

**Fig. 5 | GDP effects and amplification ratios for Rest of the World.** Results under RCP4.5-SSP2 medium sea level-rise, relative to baseline scenario, for four cases of adaptation. **a** Change in GDP. **b** Amplification ratios. Region abbreviations: NAM North America; AUZ Australia and New Zealand; ERA Eurasian countries; ECA Emerging economies- Asia; TUR Turkey; CHN China; IND India; SEA South-East Asia; LAM Latin America (w/o Venezuela); OIE Oil exporting countries (OPEC: Middle East and Africa + Venezuela); AFR Africa.

erosion, salinization, or ecosystem changes, as well as other less tangible costs such as loss of culture or biodiversity, because direct cost estimates are not readily available at global scale. One noteworthy exception is the assessment of direct economic damages due to coastal erosion by Hinkel et al.[49], but this study also indicates that the expected damages are 3 to 4 orders of magnitude smaller than those incurred by rising mean and extreme sea levels considered here. Also, potential co-benefits of planned adaptation, e.g., the creation of public space, are not included.

Finally, concerning the chosen time horizon of 2050, we emphasize that sea levels are expected to rise to much higher levels beyond 2050, especially under a high-end ice melting assumption. The high-end scenario as analyzed here reaches 1.7 m of SLR in 2100, which is higher by a factor of 3.7 compared to 2050 (0.46 m). Sea flood costs thus also continue to increase beyond 2050, while regarding the number of migrants and associated migration costs, the development over time occurs in waves, with highest amplitudes before 2050 for many regions[22]. Hence, GDP losses from the sea flood damage-channel are expected to increase further beyond 2050, however, migration costs are captured well in the results as shown in this analysis until 2050.

## Methods
### Scenarios
We make use of the RCP-SSP scenario framework. RCPs are Representative Concentration Pathways, which can be translated into

greenhouse gas emission trajectories over time[46]. The RCPs most used in the modelling community are RCP2.6, RCP4.5, RCP6.0, and RCP8.5, with, e.g., RCP4.5 representing a case with moderate GHG emissions and a forcing of 4.5 W/m² in 2100, very likely leading to 2.1–3.5 °C global mean surface temperature increase by 2100, relative to 1850–1900[50]. The RCP with strongest forcing in 2100 is RCP8.5, which would very likely lead to a mean global temperature increase between 3.3 and 5.7 °C by 2100 (relative to 1850–1900) and a likely mean SLR between 0.6 and 1.0 m by 2100 (relative to 1995–2014 sea levels) in addition to the 0.2 m increase that has already happened since 1900[50]. SSPs are Shared Socioeconomic Pathways, which represent five semi-quantitative narratives of possible socio-economic developments, which differ in terms of challenges towards mitigation and adaptation[47]. For example, SSP1 describes a sustainable development with low challenges for both mitigation and adaptation, whereas SSP3 describes a development with low growth in wealth per capita and strongly rising greenhouse gas emissions, resulting in high challenges to adaptation and mitigation. In their combination, RCPs and SSPs form different scenario worlds which can be compared to each other, e.g., to find out how an impact from climate change materializes in different futures.

From a policy makers' perspective, it is often useful to take a cautious approach and prepare for the worst-case—even if probabilities are low[51]. Hence, next to a middle-of-the road scenario, we include a possible high-end scenario (meaning the higher-end of SLR

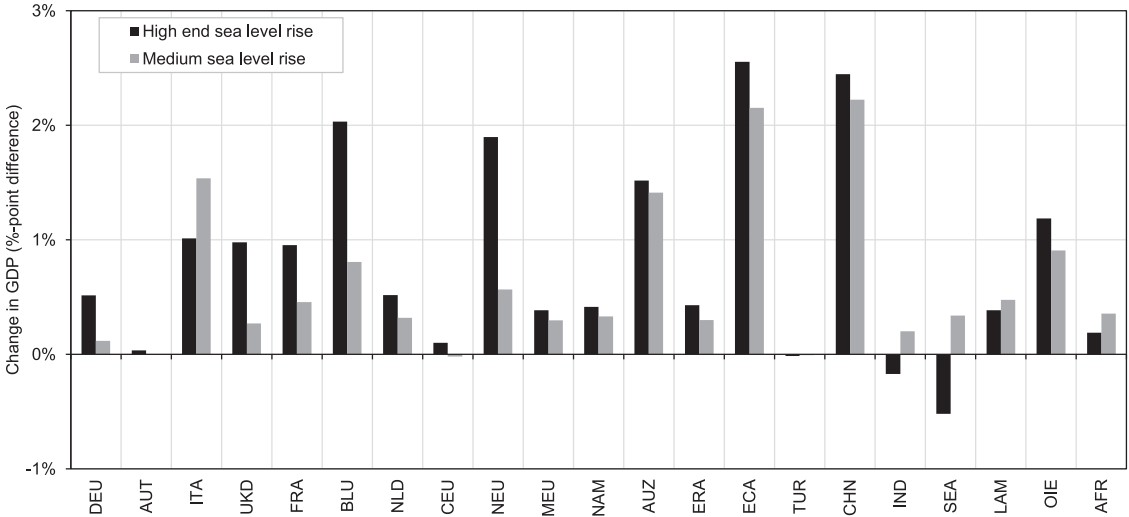

**Fig. 6 | Comparison of macroeconomic effectiveness.** Comparison between planned adaptation (sea dikes) and autonomous adaptation (migration) under RCP8.5-SSP5 in 2050 under high-end (dark grey) and medium (light grey) sea-level rise. Effectiveness is measured as %-point difference of GDP loss between planned adaptation (sea dikes) and autonomous adaptation (migration). These values are also visible as the difference between squares and diamonds in Fig. 3. Positive bars indicate that GDP losses are less severe with planned adaptation-only and vice versa. Reading example: In region ECA under high-end sea-level rise, relative GDP losses are by 2.5%-points lower (0.1% instead of 2.6%) with planned adaptation (sea dikes) than with autonomous adaptation (migration). Region abbreviations: DEU Germany; AUT Austria; ITA Italy; UKD United Kingdom; FRA France; BLU Belgium and Luxemburg; NLD Netherlands; CEU Central EU 27 + Switzerland; NEU Northern EU 27+ Liechtenstein, Norway and Iceland; MEU Mediterranean and South-eastern EU 27; NAM North America; AUZ Australia and New Zealand; ERA Eurasian countries; ECA Emerging economies- Asia; TUR Turkey; CHN China; IND India; SEA South-East Asia; LAM Latin America (w/o Venezuela); OIE Oil exporting countries (OPEC: Middle East and Africa + Venezuela); AFR Africa.

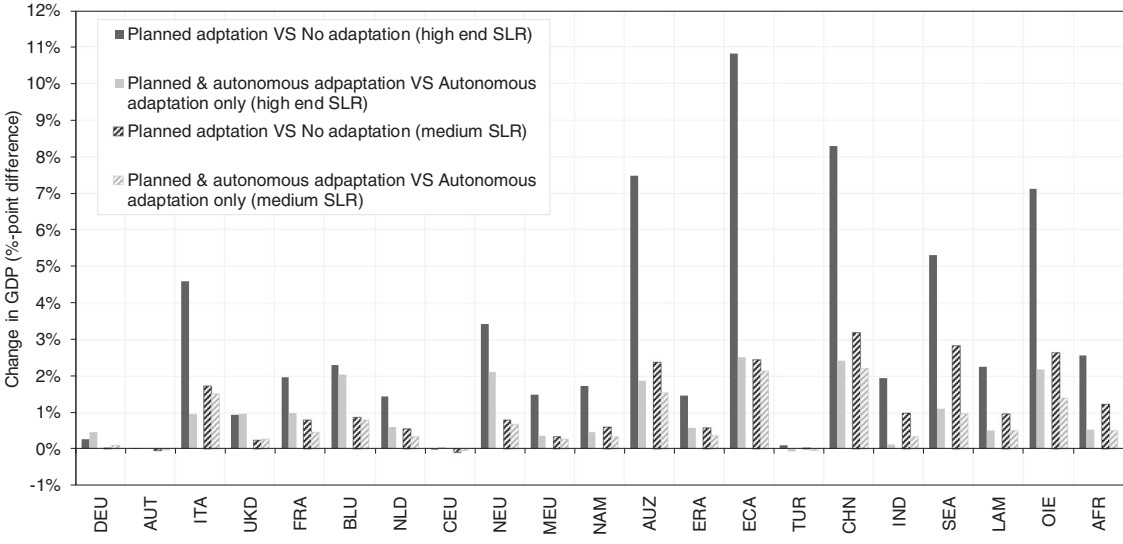

**Fig. 7 | Benefits of adaptation with different assumptions of autonomous adaptation.** Benefits for RCP8.5-SSP5 in 2050. Values are calculated as avoided relative GDP loss in %-points. Positive values indicate that GDP losses are less severe compared to the respective reference point. Solid bars show effects under high-end sea-level rise, hatched bars under medium sea-level rise. Dark grey colour shows differences between planned-adaptation-only (sea dikes) versus no-adaptation, light grey colour shows differences between planned-and-autonomous-adaptation versus autonomous-adaptation-only (migration). Reading example: in region ECA under high-end sea-level rise, planned adaptation (sea dikes) reduces GDP losses by 10.8%-points (from −10.9 to −0.1%), relative to a no adaptation scenario, but only by 2.5%-points (from −2.6 to −0.1%) when autonomous adaptation (migration) is accounted for as well. Region abbreviations: DEU Germany; AUT Austria; ITA Italy; UKD United Kingdom; FRA France; BLU Belgium and Luxemburg; NLD Netherlands; CEU Central EU 27 + Switzerland; NEU Northern EU 27+ Liechtenstein, Norway and Iceland; MEU Mediterranean and South-eastern EU 27; NAM North America; AUZ Australia and New Zealand; ERA Eurasian countries; ECA Emerging economies- Asia; TUR Turkey; CHN China; IND India; SEA South-East Asia; LAM Latin America (w/o Venezuela); OIE Oil exporting countries (OPEC: Middle East and Africa + Venezuela); AFR Africa.

projections). Specifically, we use two RCP-SSP combinations, additionally combined with different assumptions on ice melting sensitivity and adaptation. The following three scenarios are used, describing changes in sea levels relative to 2015 (SLR compared to the average of 1985–2005 are given in Supplementary Table 2).

1) RCP8.5-SSP5 with a high-end ice melting assumption with 0.39 m SLR by 2050 (1.62 m by 2100), also described as low-likelihood, high-impact storyline in the IPCC's 6th assessment report[1].
2) RCP8.5-SSP5 with medium ice melting, which corresponds to 0.19 m SLR in 2050 and 0.63 m SLR by 2100.

3)  RCP4.5-SSP2, to put the former two scenarios into perspective. This is regarded as a middle-of-the road scenario (but still not fully compatible with the Paris targets) with 0.16 m SLR by 2050 and 0.45 m SLR by 2100.

All three scenarios are run under four different adaptation cases:

a)  No adaptation: In this case, no further adaptation is assumed, meaning that current protection levels are maintained at 2015 levels, but no additional measures are taken. This means that there is neither additional protection nor any reaction in terms of autonomous retreat. Dikes with a height that fall (due to sea-level rise) below the height of the 1-in-1 year extreme water level (which is usually at the level of high tide) are permanently overtopped and disappear. Thus, the length of protected coastline gets smaller with increasing sea levels (see Supplementary Figs. 9–14). We regard this case as rather implausible and include it as a worst-case hypothetical reference point for comparison.

b)  Planned adaptation-only (sea dikes): In this case, only protective planned adaptation, financed by the public household, is implemented in the form of dikes. The local protection decision is based on local population density and local GDP per capita. As in coastal regions both population density and GDP per capita are more often increasing than decreasing, there is a slight increasing trend in the length of protected coastline over time (see Supplementary Figs. 9–14).

c)  Autonomous adaptation-only (migration): In this case the only response to SLR is the reactive autonomous retreat of people and assets from the coastline. Migration is assumed to take place when habitats fall below the water level of the 1-in-1-year event (inducing regular frequent flooding). See Supplementary Figs. 3–8 for migration numbers.

d)  Planned and autonomous adaptation: In this case planned public protection (sea dikes) is combined with reactive autonomous retreat (migration) from areas that are not protected. Only members of the population residing in areas that are not protected (due to low population density or low local GDP per capita) migrate away from the coast (if they fall below the water level of the 1-in-1-year event)

We thus span a scenario space of $3 \times 4 = 12$ scenarios. Due to substantial uncertainties in socio-economic development, we choose our time horizon until 2050, but not further. Also note that for complexity reasons, only one global circulation model (GCM HadGEM-ES2) is applied, as this projects SLR in the range of the average over multiple GCMs[36].

If not stated differently, results are given as relative change to a baseline scenario, which includes only the socio-economic development of the world (as given by the underlying SSP), but no climate change (and thus no climate change-induced sea-level rise). By comparing the climate change impact scenarios to the respective baseline, the effect of climate change can thus be isolated.

## The DIVA Model

The DIVA modelling framework[22,36] is used for assessing global coastal flood damages, protection cost, and migration. Impacts and costs are calculated for the 12,148 coastline segments defined in the DINAS-COAST database[52]. Coastal segments represent parts of the coast with homogeneous bio-physical and socio-economic characteristics.

Population exposure for each segment is obtained by overlaying Shuttle Radar Topography Mission (SRTM) elevation data[53] with Global Rural-Urban Mapping Project (GRUMP) population data[54]. Exposed population is translated into exposed assets by applying sub-national GDP per capita rates[52] to the population data, followed by applying an assets-to-GDP ratio of 2.8[55]. Extreme water level distributions are taken from the GTSR database[56] and are assumed to uniformly increase with

SLR, following 20th century observations[57]. For local relative sea level change, climate-induced SLR is complemented with glacial-isostatic adjustment[58] and delta subsidence for coastal segments associated with river deltas[59]. Estimates of current protection levels are taken from Lincke and Hinkel[11] who took protection levels for the biggest 136 coastal cities from Hallegatte et al.[55] and complemented these with expert judgement for segments not associated to one of these cities.

A protection level of zero is assumed if the population density in the 1-in-100-years floodplain is lower than 30 people per km². In unprotected areas it is assumed that no one lives below the 1-in-1-years water level in the model initialization (2015). Flood damages are calculated by combining elevation-based asset exposure with flood depths caused by extreme events and applying a depth-damage function that maps water depths into fractional damages on assets, taking into account existing protection. Expected annual flood damages are computed as the mathematical expectation of damages based on extreme event distributions[36]. Protection is modelled by hard protection infrastructure (dikes) and protection levels are forecasted by projecting the assumptions described above into the future. Cost for construction of protection infrastructure is based on national unit cost for dikes[36] and the annual maintenance cost for protection infrastructure is one percent of its capital cost. Protection infrastructure is assumed to have no protective function if the water level is higher than the height of the protection. For water levels below the height of the protection infrastructure no dike failure is possible (the dike certainly holds).

Population migrates only from coastal areas that are not protected. Specifically, population migrates when falling below the water level of the 1-in-1-year event. This assumes a rather reactive form of migration: the population is assumed to deal with rising water levels up until areas are flooded annually. With this assumption we follow literature that assumes that land is lost if it lies below the 1-in-1 year flood return level[21], which means that land is inundated on average once per year and thus generally is not usable for buildings and infrastructure. Hence the population stays as long as possible until the land is finally uninhabitable. This is supported by the literature. Empirical findings suggest that there is often an unwillingness to migrate, even under constant threat[19,60]. This unwillingness is reflected in our choice of a high flood probability threshold (1-in-1 year flood return level). The 1-in-1 year flood return level can be seen as a proxy for spring high tides.

This also indicates that the modelled retreat is not a planned or managed operation organized by a social planner, but rather an autonomous and reactive from of adaptation. Importantly, migrants are assumed to stay within their country, but move to destinations far enough inland such that they are not exposed to sea floods anymore (also in the future). Following the synthesis of Lincke and Hinkel[22], migration costs are valued at three times GDP per capita per migrant, which mirrors the costs of moving and/or rebuilding the capital stock elsewhere. Note that DIVA models real capital costs of sea floods (expected annual damages to the built environment) and does not include non-coastal zones explicitly. The economy-wide net-effect of dispersions of agglomerations at one place and of potential new agglomerations elsewhere is thus implicitly assumed to be neutral. This is supported to some extent by the literature. For example, Desmet et al.[39] find that when considering such effects of economic geography and the emergence of new agglomerations, SLR-induced GDP losses from dispersions are much weaker than without assuming new agglomerations to emerge, hence systems are rather resilient in that respect.

## The COIN-INT model

The COIN-INT model is a global, multi-regional, multi-sectoral CGE model, implemented in GAMS/MPSGE. It was originally developed as a comparative static model by Schinko et al.[61] and has been enhanced for the purpose of climate change impact assessment[62]. For the analysis

presented here, the model has been further developed; of note is that it now features recursive dynamics, i.e., the model solves in consecutive 5-year time steps and gives as a result a trajectory over time. The model is calibrated to the GTAP9 database[63], with the base year of 2011 and projects until 2050. The main characteristics are briefly summarized in this section; for details, please see the Supplementary Methods.

In COIN-INT individual countries are aggregated to 21 larger regions that share similar climatic conditions. There is a focus on Europe, hence at the European level the regional resolution is higher than for the rest of the world (ROW); see Supplementary Table 1 for details. As opposed to hard-linked IAMs (e.g., the DICE model[24,25]) the model features sectoral differentiation, which allows directly capturing indirect effects as well as sectoral winners and losers. In total there are 21 sector aggregates (see Supplementary Table 2), which are connected via input-output connections. Regarding final demand, there are two representative households in each EU region. The first is a private household that is endowed with the production factors skilled labour, unskilled labour, capital as well as natural resources (fossil resources, other natural resources, land, and $CO_2$ emission allowances). Second, in each EU region, there is also a public household, which collects taxes and provides transfers to the private household. Net-tax income is used to finance government consumption. In non-EU regions there is only one representative regional household, aggregating public, and private consumption and investment. Investment in each region is determined via a fixed savings rate (i.e., a fixed proportion of income is devoted to savings/investments). Endowments are supplied to the market and are used in combination with intermediate inputs to generate output; i.e., goods and services. Production functions of goods and services are implemented as nested constant elasticity of substitution (CES) functions.

COIN-INT is applied in its recursive dynamic version that explicitly models the pathway of economic development in 5-year time steps from 2015 to 2050. It is calibrated to nine SSP-RCP-combinations (see Supplementary Figs. 1 and 2 for $CO_2$ emissions and $CO_2$ prices, respectively). The calibration process is explained in detail in the Supplementary Methods.

### Model coupling

DIVA and COIN-INT are both calibrated to the RCP-SSP framework, hence coupling the two models is consistent. The following information from DIVA is used as input for the COIN-INT CGE model: Annual land loss due to submergence (km²/year), expected annual damages to assets by sea floods (million US\$/year), total capital stock (million US\$/year), expected annual number of people flooded per year (thousands/year), protection costs (million US\$/year), split into an investment fraction and a maintenance cost fraction, as well as migration costs (million US\$/year).

The impacts of SLR are implemented in COIN-INT via six channels:

1. **Capital costs due to sea flood damages**: Sea flood damages are implemented via a reduced capital stock, hence a reduced capital stock accumulation that leads to lower capital availability for production (i.e., a lower capital endowment in the economy). This leads to lower economic activity (as productive capacities are reduced), lower income, and in turn lower consumption and savings (subject to a fixed savings rate). Lower savings lead in turn to lower investment and thus lower capital accumulation over time (in addition to the direct effect). Reconstruction is assumed to be GDP neutral, i.e., it crowds out other generic investment. The sea flood costs to capital stock-ratio from DIVA is calculated and then applied to the capital stock accumulation equation in COIN-INT.

2. **Labour supply losses**: DIVA calculates the number of people that are flooded in each year. Following Parrado et al. (2020) we

assume that each person that is flooded within a year is not able to provide labour to the labour market for 2 out of 48 working weeks a year. We use the annual labour income per capita in each region, apply the ratio of 2/48 to it and multiply it with the number of people that are flooded to obtain the total labour costs. The resulting labour supply losses are then subtracted from the productive labour supply (endowment) in each region.

3. **Land loss**: DIVA calculates the annual land area that is lost due to SLR in each year. We cumulate this effect over time and calculate the change in land availability in each year (relative to the land area that is available in COIN-INT). The relative land loss is then implemented in COIN-INT as lower cropland availability for agricultural crop production. As the effect of land loss happens gradually, we can assume that the type of land that is close to the shore and that it not protected is land of lowest value, i.e., agricultural land.

4. **Sea dike investment costs**: Investment costs for renewing sea dikes or for upgrading them (in the case of adaptation via protection) is modelled as forced investment activity of the government agent in each model region and year. This forced investment is assumed to crowd out government consumption. Further, we assume that this investment is only effective in the short term, i.e., it has a positive effect on GDP in the year of investing (though at the cost of government consumption), but it does not build up the productive capital stock, since sea dikes cannot be regarded as a production factor that earns a rent (as opposed to other capital such as machinery or buildings). Higher sea dike investment thus leads to lower capital accumulation over time.

5. **Sea dike maintenance costs**: Maintenance costs for sea dikes are implemented as forced government consumption for construction activities, which crowds out generic government consumption.

6. **Migration costs**: Migration costs capture two aspects. First, the costs of leaving immobile assets behind, i.e., full depreciation of assets that are lost due to coastal retreat. Second, the costs of moving mobile capital away from the coastlines further inland. Both aspects fall into the broad category of capital costs and are treated as such in the CGE model, where they reduce the accumulation of productive capital (i.e., capital stock).

For a consistent flow of information across the CGE model and DIVA, all values from DIVA, expressed in US\$ PPP (Purchasing Power Parity) are converted to US\$ MER (Market Exchange Rates), the CGE model's reference, using conversion factors from the World Development Indicators[64].

### Amplification ratio

To calculate amplification ratios (ARs), we first calculate total GDP loss of a specific year in absolute terms (GDP difference between baseline and impact scenario). This GDP loss in then divided by the direct sea-level rise-induced costs of the same year, which is the sum of capital costs, labour costs, sea dike investment costs, sea dike maintenance costs as well as costs from land loss. The result measures by how much annual GDP losses are larger than annual direct costs.

### Reporting summary

Further information on research design is available in the Nature Research Reporting Summary linked to this article.

## Data availability

The macroeconomic impact data generated in this study are provided in the Source Data file (i.e., data sources underlying Figs. 1–7 and Supplementary Figs. 27–33). Data to calibrate the global CGE model COIN-INT were obtained from the Global Trade Analysis Project (GTAP

version 9, https://www.gtap.agecon.purdue.edu) as well as from the IIASA SSP database (https://tntcat.iiasa.ac.at/SspDb). Data on the direct costs of sea-level rise, as generated by the DIVA model and documented in Lincke and Hinkel[22], are deposited in a ZENODO database (https://doi.org/10.5281/zenodo.6417157). Source data are provided with this paper.

## Code availability

The COIN-INT model is implemented in GAMS/MPSGE and solved numerically using the PATH solver. The current code base of the COIN-INT CGE model developed over more than two decades at University of Graz and is not available in a publicly shareable version. The code will continue to be developed and hosted by University of Graz, Wegener Centre for Climate and Global Change (https://wegcenter.uni-graz.at/en/). Requests for code should be addressed to Gabriel Bachner.

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

## Acknowledgements

This research was made possible by support from the European Union through the project "CO-designing the Assessment of Climate CHange costs" (COACCH), funded by European Union's Horizon 2020 research and innovation programme under grant agreement number 776479 (funding was awarded to G.B., D.L., and J.H.), and the project "PROjecTing sEa-level rise: from iCe sheets to local implicaTions" (PROTECT), funded by the European Union's Horizon 2020 research and innovation programme under grant agreement 869304 (funding was awarded to D.L. and J.H.). The authors acknowledge the financial support by the University of Graz. We thank our colleagues from the Wegener Centre for Climate and Global Change, University of Graz, for feedback on earlier versions of this paper and Keith Williges for proof-reading.

## Author contributions

G.B., D.L., and J.H. designed and conducted the analysis and wrote the main paper. D.L. and J.H. prepared and analyzed the input data to the macroeconomic model, G.B. conducted the macroeconomic analysis, interpreted results, and created the figures. All authors read and commented on paper drafts and approved the final version.

## Competing interests

The authors declare no competing interests.
