## [Peer Review File · Nature Communications]

Reviewer comments, first round review –

Reviewer #1 (Remarks to the Author):

This manuscript presents an analysis of the overall economic costs of sea level rise when taking into account 1) planned protective sea walls and 2) autonomous out-migration. The authors link a model of coastal damage/adaptation with a CGE model that incorporates the long-term consequences of productive capital and labor. Results indicate that including autonomous retreat lowers the costs relative to a scenario where only protective sea walls are considered, and that the indirect effects are more consequential than direct damage to assets.

My primary concern relates to the methods employed. As written, the methods do not address where migrants go and seem to consider out-migration solely as a cost, even though the macroeconomic costs would presumably depend on the reallocation of labor and the spatial shifts in economic activity. This may just be an oversight in the description of the methods, e.g., there is an assumption about the migration destination, but the nature of that assumption seems critical to understanding the results. In-migration can of course substantially increase GDP of a region, as well as the population density (which would affect the likelihood of coastal protection, based on the assumptions used in the paper). Without adding in information about migration destinations, it is difficult for me to assess the findings.

Relatedly, the authors do not reference the paper that I believe to be most similar to the current manuscript -- Desmet et al. 2021 – Evaluating the Economic Cost of Coastal Flooding, American Economic Journal: Macroeconomics. I am not an author on that paper, and there are key differences in the goals (explicitly considering sea walls or not). However, it is clear from that paper that considering autonomous migration substantially reduces GDP loss. Of particular relevance to this paper is the notion that new clusters of economic activity emerge or are reinforced as they receive people out-migrating from flooded areas, which partially compensates for the losses in flooded areas. This seems to be a critical point that the current manuscript does not address. At the very least, the authors should contextualize their results and migration modeling with respect to this paper, given the similarities.

Second, the Discussion section asserts the benefits of using “autonomous adaptation” as a baseline scenario, but provides no discussion of the validity of their autonomous adaptation assumptions. For example, this model assumes anyone can out-migrate, despite acknowledging in the background that slow-onset climate change can trap people geographically – and SLR is precisely the type of slow-onset event that might do this. While I recognize this is a scenario-based analysis, that should be addressed in the Discussion as well as the implications for the results. This suggestion also applies to other limiting assumptions made in the modeling.

My final main concern is about the organization of the manuscript – the introductory section at the moment needs just a little bit more description of the methods before diving into the results. I recognize the core Methods come at the end, but the results at the moment are impossible to understand. The fundamental pieces of the analysis – DIVA and COIN-INT – are not even named or spelled out in full before the Results.

Minor comments:

- Abstract: not clear what “conventional” on Line 15 means in this context
- Page 3, Line 50: characterizing autonomous migration and public sea walls as extreme ends of a spectrum is confusing when there are multiple dimensions of difference (public vs. private, armoring vs. retreat, planned vs. reactive). There are private households who armor their properties, and there is reactive public adaptation. I suggest rewording.
- Page 6, Line 150: I see no mention of LTARs in the Supplementary – where are the methods?
- Page 10, Line 206: I don’t think “overestimated” is the right term here since one major difference is who pays – public or private. “Overestimated” implies that as much of the burden as possible should be borne privately.
- Page 12, Line 238: “the benefits of adaptation” – should be benefits of “public sea walls”

- The Discussion section could benefit from more attention to the distinction between public and private costs.

Reviewer #2 (Remarks to the Author):

Nature Communications: Macroeconomic Implications of High End Sea Level Rise and Coastal Migration: A Global Assessment

This is an interesting paper that considers the global impacts of sea-level rise from an economic perspective taking account of autonomous adaptation (migration in response to more frequent flooding) and/or increased protection. The authors emphasise that this is novel and in broad terms I agree with them. However, the manuscript needs substantial revision before it could be accepted,

The paper has general aspirations of informing the debate about adaptation and clearly has a multi-disciplinary team behind it. However, I hypothesise that one team member – an economist has written the manuscript with limited input from the other team members. Hence the paper has a rather jargonistic flavour being written for economists rather than a more general audience – the arguments and language should be reviewed throughout and improved for clarity for a general audience. I have indicated some cases in the detailed comments below but there are more than I can reference. The results in particular are rather dry and would greatly benefit from a revision – I have given no line by line comments here as I expect it to be rewritten.

The framing of the problem could also be much sharper and parts of the manuscript tend to ramble on with content that is correct but is not really building to a strong argument. For example, the manuscript raises the importance of international (cross-border) migration which is obviously important globally, but I am unclear if it matters in this study and there is no evidence that the migration model even considers this factor and if the IAM considers it either. So a general sharpening of the arguments is needed. There are more detailed remarks line-by-line below. Also is this the first model to conduct the analyses that are being outlined here? Richard Tol certainly included protection and migration using benefit-cost analysis in the coastal model of FUND and the paper cites some of his work – Tol (2007) – but there are other papers. So while this paper is more comprehensive than FUND is it quite as new as it claims. I also note the paper by Hinkel et al (2013) (Global and Planetary Change) which is focussed on coastal erosion rather than flooding contrasted migration with and without soft protection. Again, I suggest this is worth citing.

There should also be a focus on 2050 when the simulations end. How much sea-level rise has occurred by 2050 – probably not very much as the divergence of sea-level rise scenarios happens mainly after 2050. The paper by Lincke and Hinkel (2021) goes to 2100 and the number of migrants by 2050 is relatively small. This needs to be made clear to the reader. Is the model and these results meaningful for a much larger rise in sea level, as would happen from 2050 to 2100 under the Extreme scenario?

More detailed comments as follows.

Line 31-33 – in addition to small islands the high vulnerability and high populations of delta regions has been highlighted in most IPCC assessments.

Line 37/38 – there are many adaptation options (e.g., IPCC SROCC Report, 2019) – would be good to position the analysis as looking at selected options. As written could come across that these are the only adaptation options.

Line 43 is Nerem et al (2018) the correct reference here – it is about acceleration in sea-level rise and not about migration?

Line 43-46 – narrative is not so clear – comparing apples and oranges and maybe pears – different hazards and different assumptions – the point is that assuming no adaptation displaced numbers are large and even considering benefit-cost protection the numbers remain significant. Examples should all be SLR. If you want to discuss migration due to other causes suggest that is in a separate paragraph.

Line 49 – example of “autonomous adaptation”

Line 53 “We regard these two possibilities as extreme ends of the adaptation spectrum, with e.g. managed retreat in-between (Alexander et al., 2012; Hino et al., 2017; Kousky, 2014).” Not sure you can say this defines the extremes – adaptation is more diverse than this – tying into my earlier comment of positioning this work in the wider coastal adaptation literature. For example

what about advance seaward and build vertically up – or flood warnings and preparedness?
Line 52-57 rather rambling text -- this could be moved to the more structure discussion about adaptation.

Line 58 – why does it matter if migrants cross a border or not – the question here is do they move? As far as I can see this is a red herring.

Line 58-59 Takes a very narrow view of migration, which covers a spectrum of behaviours – further most migration today is within country and with SLR will that change – unlikely. So there seem to be lots of unsubstantiated and incorrect assumptions here. Recommend that the authors look at the papers on empirical data (e.g., Adger et al., 2021; <https://doi.org/10.1016/j.oneear.2020.12.009>). Later text (to line 69) is better but again I would say rambles a little.

Line 57 – “individuals” – what about households?

Line 70-79 Add the FUND model here as it has been applied to these SLR questions quite a few times and more times than cited in this paper.

Line 85/86 “any way” ?? I think trying to same in “some way” and no adaptation does not exist. Are there not other papers than Hinkel et al (2014) that substantiate this view.

Line 86/88 – there is extensive adaptation infrastructure in many developing countries such as the delta countries of Asia (e.g., Bangladesh, Thailand, Vietnam, China). Here as far as I am aware the defences are being upgraded – not necessarily for SLR but to raise the standard and quality. So I think this statement needs more thought as to where they mean exactly.

Line 89/90 – does not describe the model adequately as it includes protection as well.

Line 94-101 – Scenarios are methods not results.

Line 94-101 – All the results end in 2050 so the sea-level rise in 2100 is of little practical value – report the rise in sea level by 2050 as well – which will be much smaller. As this scenario is not defined, especially for the Extreme case it is hard to assess the paper and its results.

Line 220 “costal”??

Line 223 “other impacts” – what other impacts?

Line 228/229 “planned adaptation, as the latter needs high (public) investments into protective infrastructure which binds capital that might be used more productively elsewhere in the economy.” Implies that public investment is protection – but public investment can go into other measures.

Line 233 “The here taken perspective is a rather narrow one,” – poorly formulated

Line 239 – there are many forms of autonomous adaptation so the case provided as a reference is simply an example and should be stated as such.

Line 242 “dynamics” – what dynamics – rather unclear

Line 381 – change in temperature relative to what baseline? This comment is true throughout the methods.

Line 388 – focusses or includes?

Line 392-397 – as the analysis focusses on 2050, I am much more interested in the changes to 2050 – please add in all cases.

Line 398-400 – number and define the four adaptation scenarios

Line 406 – is non-climate-change induced SLR included and if so how?

Line 420-422 – protection levels are all pretty uncertain – to reflect this add “Estimates of

Line 424 “In unprotected areas it is assumed that nobody lives below the 1-in-1-years water level.” What is the base year?

Line 428-430 --- what damage function is assumed for dike failure – this might have a significant influence on the costs.

Line 434 “flooded regularly.” – restate the threshold.

Line 432-438 – does where they migrate to have any effect on the results – presumably they are assumed to migrate out of the coastal zone (how defined?) – this needs to be added. Are there any implications of destination on the integrated assessment and where capital accumulates? Also the authors raises international migration as important earlier – I disagree – but does this model say anything about this – seemingly not?

Line 450 – add a source to an example hard-wired IAM.

Line 450 to “capture)

Line 466-471 – what about number of migrants per year – a useful diagnostic parameter?

Line 481 – Labour costs – where does this assumption come from –seems essentially made up. On what basis and is there any empirical evidence to support. Fair enough if it is a first attempt to make the link but some reflection here is needed.

Supplemental Material – select and add some diagnostic results for the DIVA runs – to better understand all the economic results it would be good to have some real world properties of each simulation – parameters that spring to mind are length of coast protected and number of migrants. The authors may have additional ideas. These might be referred to in the main manuscript.

Reviewer #1:

Comment by reviewer	Response by authors
This manuscript presents an analysis of the overall economic costs of sea level rise when taking into account 1) planned protective sea walls and 2) autonomous out-migration. The authors link a model of coastal damage/adaptation with a CGE model that incorporates the long-term consequences of productive capital and labor. Results indicate that including autonomous retreat lowers the costs relative to a scenario where only protective sea walls are considered, and that the indirect effects are more consequential than direct damage to assets.	We sincerely thank this reviewer for the important remarks and questions. We have re-written most of the manuscript. In the provided (extra) tracked change-version of the manuscript we thus omitted deletions for the sake of readability and only indicate new text. We believe that the new version has considerably improved over the initial one and hope to have addressed all issues satisfactorily.
My primary concern relates to the methods employed. As written, the methods do not address where migrants go and seem to consider out-migration solely as a cost, even though the macroeconomic costs would presumably depend on the reallocation of labor and the spatial shifts in economic activity. This may just be an oversight in the description of the methods, e.g., there is an assumption about the migration destination, but the nature of that assumption seems critical to understanding the results. In-migration can of course substantially increase GDP of a region, as well as the population density (which would affect the likelihood of coastal protection, based on the assumptions used in the paper). Without adding in information about migration destinations, it is difficult for me to assess the findings. Relatedly, the authors do not reference the paper that I believe to be most similar to the current manuscript -- Desmet et al. 2021 – Evaluating the Economic Cost of Coastal Flooding, American Economic Journal: Macroeconomics. I am not an author on that paper, and there are key differences in the goals (explicitly considering sea walls or not). However, it is clear from that paper that considering autonomous migration substantially reduces GDP loss. Of particular relevance to this paper is the notion that new clusters of economic activity emerge or are reinforced as they receive people out-migrating from flooded areas, which partially compensates for the losses in flooded areas. This seems to be a critical point that the current manuscript does not address. At the very least, the authors should contextualize their results and migration modeling with respect to this paper, given the similarities.	We have added more information in the main body of the text as well as in the methods section to better describe and motivate the assumptions with respect to migration, as well as regarding the type of costs that we include. To summarize: We assume that migrants are rather resistant to flooding and accommodate for a long time. They only leave coastal zones as soon as they are flooded annually and move to safe zones further inland, where no further risk of flooding exists. No international migration is assumed (an assumption that reviewer #2 pointed out to be realistic and which we also support by referring to the literature). We now also refer to Desmet et al., a very interesting paper, several times. We clearly have a different focus, though. Our method captures capital stock damages (next to other costs) from sudden-onset sea floods. Desmet et al argue that capital costs are low as natural depreciation is slower than SLR, which is however only valid for slow-onset SLR. Again, referring to Desmet et al., we acknowledge that we do not include costs associated with the dispersion of economic agglomerations as our model is not capable to do so. Hence, implicitly assume that the net effect from dispersion and new emergence of agglomerations is neutral. For an explicit capturing of such effects a spatial model would be needed, which is out of scope of our analysis.
Second, the Discussion section asserts the benefits of using “autonomous adaptation” as a baseline scenario, but provides no discussion of the validity of their autonomous adaptation assumptions. For example, this model assumes anyone can out-migrate, despite acknowledging in the background that slow-onset climate change can trap people geographically – and SLR is precisely the type of slow-onset event that might do this. While I recognize this is a scenario-based analysis, that should be addressed in the Discussion as well as the implications for the results. This suggestion also applies to other limiting assumptions made in the modeling.	We now better motivate and validate our scenario assumptions with respect to migration in the methods section (DIVA model description). In a nutshell, we argue – supported by the literature – that it is more plausible to assume that people will leave their homes autonomously, when being flooded annually (this is only the case in unprotected areas with low population densities), than staying and accepting regular losses without any further private adaptation (which is the usually adopted assumption). Also other limiting assumptions and caveats of our study are now discussed in more detail in the discussion section.

My final main concern is about the organization of the manuscript – the introductory section at the moment needs just a little bit more description of the methods before diving into the results. I recognize the core Methods come at the end, but the results at the moment are impossible to understand. The fundamental pieces of the analysis – DIVA and COIN-INT – are not even named or spelled out in full before the Results.	We have restructured the whole manuscript such that it can be read smoothly from the beginning to the end.
 • Abstract: not clear what “conventional” on Line 15 means in this context 	We rephrased.
 • Page 3, Line 50: characterizing autonomous migration and public sea walls as extreme ends of a spectrum is confusing when there are multiple dimensions of difference (public vs. private, armoring vs. retreat, planned vs. reactive). There are private households who armor their properties, and there is reactive public adaptation. I suggest rewording. 	We rephrased and added on the multiple dimensions of adaptation.
 • Page 6, Line 150: I see no mention of LTARs in the Supplementary – where are the methods? 	We only use the abbreviation “AR” for amplification ratio now.
 • Page 10, Line 206: I don’t think “overestimated” is the right term here since one major difference is who pays – public or private. “Overestimated” implies that as much of the burden as possible should be borne privately. 	We rephrased. As we measure the economy-wide costs/benefits in terms of GDP, we do not differentiate between private and public in that respect. Yet, we rephrased the section, as “overestimate” is in fact not the right term. Also, taking a non-market perspective, the “price” of autonomous adaptation via migration might be higher than given in this analysis, since it also is about the loss of non-market values (e.g. trauma, loss of culture...). We take this up in the discussion section.
 • Page 12, Line 238: “the benefits of adaptation” – should be benefits of “public sea walls” 	We rephrased.
 • The Discussion section could benefit from more attention to the distinction between public and private costs. 	We added text in the discussion section on the differences of cost burdens between private and public actors. We also now discuss other types of non-tangible costs, which are important but not included in our assessment.

Reviewer #2:

Comment by reviewer	Response by authors
This is an interesting paper that considers the global impacts of sea-level rise from an economic perspective taking account of autonomous adaptation (migration in response to more frequent flooding) and/or increased protection. The authors emphasise that this is novel and in broad terms I agree with them. However, the manuscript needs substantial revision before it could be accepted,	We sincerely thank this reviewer for the important remarks and questions. We have re-written most of the manuscript (which was demanded specifically for the results section). In the provided (extra) tracked change-version of the manuscript we thus omitted deletions for the sake of readability and only indicate new text. We believe that the new version has considerably improved over the initial one and hope to have addressed all issues satisfactorily.
The paper has general aspirations of informing the debate about adaptation and clearly has a multi-disciplinary team behind it. However, I hypothesise that one team member – an economist has written the manuscript with limited input from the other team members. Hence the paper has a rather jargonistic flavour being written for economists rather than a more general audience – the arguments and language should be reviewed throughout and improved for clarity for a general audience. I have indicated some cases in the detailed comments below but there are more than I can reference in my review.	The team of authors has re-written most parts of the manuscript to avoid too much economic jargon. Yet, as it is clearly an economic analysis, some economic language cannot be avoided.
The results in particular are rather dry and would greatly benefit from a revision – I have given no line by line comments here as I expect it to be rewritten.	We have rewritten the whole section.
The framing of the problem could also be much sharper and parts of the manuscript tend to ramble on with content that is correct but is not really building to a strong argument. For example, the manuscript raises the importance of international (cross-border) migration which is obviously important globally, but I am unclear if it matters in this study and there is no evidence that the migration model even considers this factor and if the IAM considers it either. So a general sharpening of the arguments is needed. There are more detailed remarks line-by-line below.	We have now sharpened our story that we want to tell as well as the arguments. We see two major contributions. First, the methodological improvement with respect to the inclusion of autonomous adaptation, and second, the insights with respect to the economic consequences of different adaptation strategies.
One specific limitation that needs to be stated is that it is only a partial analysis of sea-level rise risks – the paper focuses on coastal flooding and inundation. It does not consider erosion or salinisation or ecosystem changes which are other significant impacts of sea-level rise as described in all the IPCC assessments. This limitation is reasonable but needs to be mentioned explicitly.	We have included these issues in the discussion section and put them into perspective.
Also is this the first model to conduct the analyses that are being outlined here? Richard Tol certainly included protection and migration using benefit-cost analysis in the coastal module of FUND and the paper cites some of his work – Tol (2007) – but there are several other papers with a strong coastal focus. So while this paper is more comprehensive than FUND is it quite as new as it claims?	We now better describe the novelty of our assessment and the drawbacks of the existing literature. The work by Richard Tol and the FUND model is certainly important and we have included more references to this stream of work. The main issue that we see with the existing studies is that the capture migration only very stylized in the form of optimal forward-looking planning and also not triggered by coastal flooding but rather by slow-onset SLR.
I also note the paper by Hinkel et al (2013) (Global and Planetary Change) which is	We now cite this paper.

focussed on coastal erosion rather than flooding contrasted migration with and without soft protection. Again, I suggest this paper is worth citing.	
There should also be a focus on 2050 when the simulations end. How much sea-level rise has occurred by 2050 – probably not very much as the divergence of sea-level rise scenarios happens mainly after 2050. The paper by Lincke and Hinkel (2021) goes to 2100 and the number of migrants by 2050 is relatively small. This needs to be made clear to the reader. Is the model and these results meaningful for a much larger rise in sea level, as would happen from 2050 to 2100 under the Extreme scenario?	We now clearly focus on 2050 and give all relevant indicators also for this year. Of course, SLR is expected to reach much higher levels beyond 2050, with potentially much higher damage costs and GDP losses from sea floods. With respect to migration numbers and associated costs, we already see peaks in the 2030s and 2040s (Supplementary Figures 3-8), hence the time horizon until 2050 captures much of these effects already.
The focus on 2050 really makes me question the title – “High End” sounds dramatic and impressive but when I read the title, I was expecting a longer-term perspective. High End sea-level rise in 2050 is not very large as I have already said. Hence I would recommend changing the title to something like: “Macroeconomic Implications of Sea Level Rise, Flooding and Coastal Migration: A Global Assessment” – this means the limitation I mentioned earlier is also explicit and the long title is shortened by one word.	We now emphasize in the introduction that ice melting-sensitivity is very relevant already in the first half of the 21st century. SLR in 2050 is by a factor 2 higher when assuming high-end ice melting as compared to a medium ice-melting assumption. We still believe that we can call this “high-end” scenario, as we are on a high-end trajectory, with also significantly higher SLR in 2050 and 2100. Also when looking at the latest IPCC report, already in 2050, SLR is larger by a factor of 1.4 in the IPCCs high-end scenario, than in the central scenario (+0.56m versus +0.39m in 2050, both within RCP8.5-SSP5). Nevertheless we have decided to change the title into: “The macroeconomic effects of adapting to high-end sea-level rise via protection and migration”
More detailed comments as follows. Line 31-33 – in addition to small islands the high vulnerability and high populations of delta regions has been highlighted in most IPCC assessments.	We added.
Line 37/38 – there are many adaptation options (e.g., IPCC SROCC Report, 2019) – it would be good to present the analysis as looking at selected options from the full menu. As written could come across that these are the only adaptation options.	We now explain the many dimensions of adaptation to sea-level rise in the introduction more comprehensively and make clear that we analyze two specific options from the full menu (protection and retreat).
Line 43 is Nerem et al (2018) the correct reference here – it is about acceleration in sea-level rise and not about migration?	We deleted this reference.
Line 43-46 – narrative is not so clear – comparing apples and oranges and maybe pears – different hazards and different assumptions – the point is that assuming no adaptation displaced numbers are large and even considering benefit-cost protection the numbers remain significant. Examples should all be due to SLR. If you consider it necessary to discuss migration due to other causes suggest that is in a separate paragraph.	We rephrased and do not put so much weight on international migration anymore, as it is not relevant for this analysis.

Line 49 – it is an ‘example of’ “autonomous adaptation”	We rephrased.
Line 53 “We regard these two possibilities as extreme ends of the adaptation spectrum, with e.g. managed retreat in-between (Alexander et al., 2012; Hino et al., 2017; Kousky, 2014).” Not sure you can say that this defines the extremes – adaptation has several dimensions so extremes in what sense? – tying into my earlier comment of positioning this work in the wider coastal adaptation literature. For example what about advance seaward and build vertically up – or flood warnings and preparedness?	We rephrased and made clear that more options than the ones analyzed here would be possible.
Line 52-57 rather rambling text -- this could be moved to the more structured discussion about adaptation mentioned above.	We rephrased and took up the issue of nature based solution in the introduction where we present the full set of adaptation categories.
Line 58 – why does it matter if migrants cross a border or not – the question here is do they move? As far as I can see this is a red herring.	We rephrased to be clearer. Our analysis does not include international migration.
Line 58-59 Takes a very narrow view of migration, which covers a spectrum of behaviours and is already a fundamental characteristic of human systems such as rural to urban migration – further most migration today is within country and with SLR will that change – unlikely. So there seem to be lots of unsubstantiated and incorrect assumptions here. Recommend that the authors look at papers on empirical data on this topic. Later text (to line 69) is better but again I would say rambles a little.	We have rephrased and now include more empirical literature on that issue, which supports our assumption. International migration is not such an important issue in this context, and we now clearly state that the included migration costs stem from moving capital assets further inland (but within country borders).
Line 57 – “individuals” – what about households?	We rephrased.
Line 70-79 Add the FUND model here as it has been applied to these SLR questions quite a few times and more times than cited in this paper.	We added the FUND model and references.
Line 85/86 “any way” ?? I think trying to same in “some way” and no adaptation does not exist. Are there not other papers than Hinkel et al (2014) that substantiate this view as this is a key point where linkage to earlier studies is needed.	We now better back this up with more references.
Line 86/88 – there is extensive adaptation infrastructure in many developing countries such as the delta countries of Asia (e.g., Bangladesh, Thailand, Vietnam, China). Here as far as I am aware the defences are being upgraded – not necessarily for SLR but to raise the standard and quality. So I think this statement needs more thought as to where in the world they mean exactly.	We rephrased. What we actually wanted to say here is that also other adaptation responses than protection should be included for comparison of cost-effectiveness, particularly retreat as it is the other major adaptation response to be expected.

Line 89/90 – does not describe the model adequately as it includes protection as well.	We rephrased.
Line 94-101 – Scenarios are methods not results. Restructure paper.	We restructured the paper.
Line 94-101 – All the results end in 2050 so the sea-level rise in 2100 is of little practical value – report the rise in sea level by 2050 as well – which will be much smaller. As this scenario is not defined, especially for the Extreme case it is currently quite hard to assess the paper and its results.	We added all necessary information also for 2050.
Line 220 “costal”??	Corrected.
Line 223 “other impacts” – what other impacts?	Not relevant any more after rewriting.
Line 228/229 “planned adaptation, as the latter needs high (public) investments into protective infrastructure which binds capital that might be used more productively elsewhere in the economy.” Implies that public investment is protection – but public investment can and does go into other measures such as land use planning, building standards and codes, flood warning systems and emergency planning, etc. These may be less costly but to be effective they require sustained investment.	We rephrased to make clear that we mean investments in protection infrastructure that binds capital in an unproductive way. We also took up the issue of different forms of public adaptation and respective trade-offs in the discussion.
Line 233 “The here taken perspective is a rather narrow one,” – poorly formulated	We rephrased.
Line 239 – there are many forms of autonomous adaptation so the case provided as a reference is simply an example and should be stated as such.	We rephrased.
Line 242 “dynamics” – what dynamics – this is almost meaningless for a general readership.	We added a clearer description of what we mean.
Line 381 – change in temperature relative to what baseline? The baseline is often unclear throughout the methods so review and revise throughout.	We added information.
Line 388 – focusses or includes?	Changed to “included”
Line 392-397 – as the analysis focusses on 2050, I am much more interested in the changes to 2050 – please add this information in all cases.	We added information.

Line 398-400 – number and define the four adaptation scenarios in more detail	We added numbers and text for clear definitions of the adaptation cases.
Line 406 – is non-climate-change induced SLR included and if so how?	Yes, due to subsidence. What enters the CGE model is only the climate CHANGE-induced part of SLR, though. Put differently, also in the baseline there are changes in sea levels.
Line 420-422 – protection levels are all pretty uncertain – to reflect this and add “Estimates of	Added “Estimates of...”
Line 424 “In unprotected areas it is assumed that nobody lives below the 1-in-1-years water level.” What is the base year?	We added information.
Line 428-430 --- what damage function is assumed for dike failure – this will have a significant influence on the impacts and costs across the modelling as shown by Hinkel et al (2021) in Earth’s Future..	We added information.
Line 434 “flooded regularly.” – restate the threshold.	Replaced “regularly” by “annually”
Line 432-438 – does where they migrate to have any effect on the results – presumably they are assumed to migrate out of the coastal zone (how defined?) – these assumptions need to be stated. Are there any implications of destination on the integrated assessment and where capital accumulates? If people migrated short distances presumably coastal investment would be higher so this has a large influence on the results. So this is quite a significant point that is not addressed. Please make sure the revised manuscript addresses this point. Also the authors raises international migration as important earlier – I disagree – but does this model say anything about this – seemingly not?	We added information on that. The assumption is, that they migrate further inland into a safe zone but remain within the country. So, there is no SLR induced international migration assumed and therefore capital accumulation does not change due to the migration process itself (but of course due to the incurred costs and changes in economic activity). We explain in more detail now.
Line 450 – add at least one source to an example of a hard-wired IAM.	Added.
Line 450 to “capture)	Corrected.
Line 466-471 – what about number of migrants per year – this is a useful diagnostic parameter for a general readership?	As indicated, we do not directly use the number of migrants per year in COIN-INT, but only the estimated costs of migration in terms of a form of capital costs. As there is no international migration assumed, there are no effects in terms of labor supply or GDP from this channel. We now give number of migrants per year in the Supplementary Figures.
Line 481 – Labour costs – where does this assumption come from –seems essentially invented.. On what basis and is there any empirical evidence to support. Fair enough if it is a first attempt to make the link but some reflection here is needed. How sensitive is the	As indicated, this assumption is taken over from Parrado et al. (2020). We agree that this assumption is not supported by empirical evidence. As mentioned in Parrado et al. (2020) the effect on GDP from this channel is

model to this assumption?

rather limited though and thus we decided to keep it as it is.

To be on the safe side, we did a sensitivity analysis and switched off this particular impact channel, indicating that the contribution of this channel to the total GDP effect is negligible.
We show this in the supplementary information.

Supplemental Material – select and add some diagnostic results for the DIVA runs – to better understand all the economic results it would be good to have some real world properties of each simulation – parameters that spring to mind are length of coast protected and number of migrants. The authors may have additional ideas. These might be referred to in the main manuscript.

We created additional plots and added them in the supplementary material. We took up the suggestions by the reviewer and show number of migrants per year as well as the length of protected coasts. We refer to these metrics in the Methods section.

Reviewer comments, second round review –

Reviewer #1 (Remarks to the Author):

Overall, the manuscript has been substantially revised, and the revision is much clearer than the initial submission. There are several areas where the contribution of this analysis can be made more clear, which I highlight below. These concerns are relatively minor.

- Page 2: “low adaptation capacities” and “vulnerable” both need to be defined if they are being used to classify countries; lots of Global South communities are highly adaptive even though they have less financial resources
- Page 3: it would be helpful to actually describe some of the findings of the most relevant past work here, instead of just citing that it exists.
- Page 3, limitations paragraph: this manuscript still does use a “no adaptation” scenario, just as the others. To me, the point being made here is that changing the baseline to assume autonomous adaptation makes a big difference to assessing the effectiveness of different responses (Figure 7). That is valuable, but then the contribution in this paper is about testing the importance of including autonomous adaptation in baseline scenarios. Suggest rephrasing.
- Page 3, limitations paragraph: it is claimed that adaptation scenarios considered focuses exclusively on coastal protection, but that statement seems too strong – considering the FUND work that is noted later on, as well as multiple papers that are referenced in the manuscript that consider migration/retreat – Diaz 2016 and Desmet et al. The difference between social optimal planning for retreat + reactive migration is important and worth highlighting.
- Page 5: I do not think it is clear that “autonomous adaptation will happen in any case”, especially not under the assumptions made in this manuscript – that anyone can migrate and that it will be after hitting an annual flood exposure threshold. There are people living below the high-tide line today (see Hauer et al., 2021 for a US-focused analysis of this), and there are people who will face significant constraints to moving. We have very little empirical evidence of the link between migration and SLR to date, so parameterizing “autonomous migration” would be very difficult
- Page 11, “...ice-melting uncertainty is also the dominating uncertainty at a macroeconomic level” important to note this is only through 2050
- Figure 3: is the cost for the autonomous adaptation scenario larger than for no adaptation for DEU?
- Page 16 (bottom para) – see comment about Page 5 autonomous migration assumption
- Page 17: another limitation worth noting is the assumptions that were not sensitivity-tested, including assumptions about what places will be protected (based on a static population density), and assumptions about the cost of migration

Reviewer #2 (Remarks to the Author):

The authors have done a good job of responding to the reviews and I am broadly happy with the paper and seeing it published. Before final acceptance I have a few points that should be addressed as they will make the paper more robust.

1. Reference 22 takes a quite stylised view of migration with SLR as the driver of adaptation. However, in the real world, migration is much more complex and the paper cannot say “To avoid this complexity, we focus on modelling out-migration due to SLR only, which is the dominant push factor when it comes to human mobility in the coastal floodplain²²”. To give an example, the analysis of Jamero et al (2017) shows people NOT migrating due to relative sea-level rise and there are many other papers should the complexity of these responses. Given the focus in the paper under review on 2050 and relatively small rises in sea level this point is even stronger. Simply change to “To avoid this complexity, we focus on modelling out-migration due to SLR only, following Lincke and Hinkel²²”. This avoids a speculative statement that is not required. There is no need to add this reference to your citations.
2. The sea-level rise from 2015 to 2050 under each scenario should be explicitly stated in the main

body of the paper as there is confusion over the base years. In the methods it is stated as 1985 to 2005. This means that the SLR over the period of 2015 to 2050 is smaller than the numbers reported in the paper.

3. "upgrading coastal protection as response to local SLR for centuries" – change to "upgrading coastal protection as response to local SLR and other coastal hazards for centuries" – historic adaptation is not just a response to local SLR.

4. "We acknowledge that some of the IAM literature based on the FUND model has considered retreat 12,36, but only as response to mean SLR and not extremes," – to be fair to FUND this is not quite correct – I would say "FUND uses stylised damage functions based on fairly crude data as extremes are implicit. Damage as a function of mean SLR is not completely wrong if extremes are embedded in it. But I agree that the data in this paper is better and resolves more detail of the processes.

5. "We embed our analysis in the RCP-SSP framework: Specifically, the following three scenarios are analysed until 2050." – is the AR6 the source – reference needed here rather than wait until the Methods.

6. "relative to current sea levels, i.e., 1995-2014" – these are NOT current sea levels – they are "relative to 1995-2014 sea levels".

If these changes are made, I am happy for the paper to be published. I can look at the paper if needed or this can be done by the editor.

References

Laurice Jamero, M., Onuki, M., Esteban, M. et al. Small-island communities in the Philippines prefer local measures to relocation in response to sea-level rise. *Nature Clim Change* 7, 581–586 (2017). <https://doi.org/10.1038/nclimate3344>

We thank again the two anonymous reviewers for their valuable feedback and hope to have addressed all remaining issues satisfactorily. You can find our 1:1 response below.

Reviewer #1 (Remarks to the Author):

Overall, the manuscript has been substantially revised, and the revision is much clearer than the initial submission. There are several areas where the contribution of this analysis can be made more clear, which I highlight below. These concerns are relatively minor.

Comment by R#1	Response
Page 2: “low adaptation capacities” and “vulnerable” both need to be defined if they are being used to classify countries; lots of Global South communities are highly adaptive even though they have less financial resources	Thank you. It is correct, that this was still too blurry. What should be at focus is risk, which is the intersection of hazard, exposure and vulnerability, with the latter including adaptive capacity (besides other factors). We have rephrased and now use the term “risk” instead.
Page 3: it would be helpful to actually describe some of the findings of the most relevant past work here, instead of just citing that it exists.	We are now going into more detail on the results of the few macroeconomic studies which include migration in the context of SLR. We discuss their findings together with the respective methodological drawbacks on p.3-4.
Page 3, limitations paragraph: this manuscript still does use a “no adaptation” scenario , just as the others. To me, the point being made here is that changing the baseline to assume autonomous adaptation makes a big difference to assessing the effectiveness of different responses (Figure 7). That is valuable, but then the contribution in this paper is about testing the importance of including autonomous adaptation in baseline scenarios. Suggest rephrasing.	Thanks for pointing this out. We have rephrased the paragraph following the limitations of existing literature to be clearer on our contribution and also included this aspect in the abstract.
Page 3, limitations paragraph: it is claimed that adaptation scenarios considered focuses exclusively on coastal protection, but that statement seems too strong – considering the FUND work that is noted later on, as well as multiple papers that are referenced in the manuscript that consider migration/retreat – Diaz 2016 and Desmet et al. The difference between social optimal planning for retreat + reactive migration is important and worth highlighting.	This is correct and we realize that we have not been fully precise. We now have changed the text to describe in greater detail how other macroeconomic studies have included SLR-induced migration, their respective weaknesses and key findings. In regard to FUND-based work we now explicitly state the difference between planned (perfect foresight) and reactive migration. Diaz is a global analysis on least cost direct adaptation costs, but not a macroeconomic study. Please also see our answer to the comment on describing also the findings of the most relevant past work.

Comment by R#1	Response
Page 5: I do not think it is clear that “autonomous adaptation will happen in any case”, especially not under the assumptions made in this manuscript – that anyone can migrate and that it will be after hitting an annual flood exposure threshold. There are people living below the high-tide line today (see Hauer et al., 2021 for a US-focused analysis of this), and there are people who will face significant constraints to moving. We have very little empirical evidence of the link between migration and SLR to date, so parameterizing “autonomous migration” would be very difficult	We agree with the reviewer that current evidence does not allow to parameterize migration in any robust way. Hence, the only thing one can do is explore plausible assumptions, which we now state in the discussion. Furthermore, we now argue that our assumption that autonomous adaptation occurs is much more realistic than a scenario without any adaptation response. But we also emphasize that our assumption is only an example of one form of autonomous adaptation, which becomes clearer now throughout the text.
Page 11, “...ice-melting uncertainty is also the dominating uncertainty at a macroeconomic level” important to note this is only through 2050	We have added this information.
Figure 3: is the cost for the autonomous adaptation scenario larger than for no adaptation for DEU?	Yes. This is due to international trade effects, triggered by SLR. In fact, there are international spillovers across all regions, and DEU can benefit from lower import prices for certain goods (particularly energy) which increases DEU’s productivity. However, DEU also suffers SLR-induced direct costs at its own coastlines as well as from higher import prices for some other goods. The given results already show the “net” effect of all these interactions. In general, these spillover effects are weaker in the adaptation scenarios as other countries’ damages are being reduced strongly. However, the magnitudes of the individual effects for individual regions depends on many factors, such as their specific sectoral and regional trading partners and also how the different types of adaptation translate into different effects on international product prices. For the autonomous-adaptation-only scenario we have the following situation in DEU: Even though the direct damage to DEU (capital loss etc.) are reduced by adaptation, the “loss” of positive spillover effects in combination with the residual direct effects is more severe than the GDP losses in the no-adaptation case.
Page 16 (bottom para) – see comment about Page 5 autonomous migration assumption	Please see our response to the comment about p.5.

Comment by R#1	Response
Page 17: another limitation worth noting is the assumptions that were not sensitivity-tested, including assumptions about what places will be protected (based on a static population density), and assumptions about the cost of migration	We agree that these limitations are worth mentioning and we have done so. Regarding assumptions on migration unit cost and the autonomous migration flood risk threshold we looked at this in another paper and refer to this paper now in the manuscript. Regarding the per capita migration costs, we have carried out additional model runs with the CGE model to test the sensitivity when halving and doubling the default assumption. We find results to be robust and mention this now in the section "Sensitivity of impacts" and give additional results in the Supplementary Material (Supplementary Figures 32 and 33).

Reviewer #2 (Remarks to the Author):

The authors have done a good job of responding to the reviews and I am broadly happy with the paper and seeing it published. Before final acceptance I have a few points that should be addressed as they will make the paper more robust.

Comment by R#2	Response
1. Reference 22 takes a quite stylised view of migration with SLR as the driver of adaptation. However, in the real world, migration is much more complex and the paper cannot say “To avoid this complexity, we focus on modelling out-migration due to SLR only, which is the dominant push factor when it comes to human mobility in the coastal floodplain ²² ”. To give an example, the analysis of Jamero et al (2017) shows people NOT migrating due to relative sea-level rise and there are many other papers should the complexity of these responses. Given the focus in the paper under review on 2050 and relatively small rises in sea level this point is even stronger. Simply change to “To avoid this complexity, we focus on modelling out-migration due to SLR only, following Lincke and Hinkel ²² ”. This avoids a speculative statement that is not required. There is no need to add this reference to your citations.	Thanks for raising this issue and for providing already a solution. We have change according to the suggestion of the reviewer.
2. The sea-level rise from 2015 to 2050 under each scenario should be explicitly stated in the main body of the paper as there is confusion over the base years. In the methods it is stated as 1985 to 2005. This means that the SLR over the period of 2015 to 2050 is smaller than the numbers reported in the paper.	Thank you for flagging this inconsistency. We now report all numbers of SLR relative to 2015 (and thus somewhat lower numbers). In addition, we added a table in the Supplementary Material with the reference period of 1985-2005.
3. “upgrading coastal protection as response to local SLR for centuries” – change to “upgrading coastal protection as response to local SLR and other coastal hazards for centuries” – historic adaptation is not just a response to local SLR.	We have change according to the suggestion of the reviewer.
4. “We acknowledge that some of the IAM literature based on the FUND model has considered retreat 12,36, but only as response to mean SLR and not extremes,” – to be fair to FUND this is not quite correct – I would say “FUND uses stylised damage functions based on fairly crude data as extremes are implicit. Damage as a function of mean SLR is not completely wrong if extremes are embedded in it. But I agree that the data in this paper is better and resolves more detail of the processes.	We have rephrased and also added more information on the few macroeconomic papers that include migration.

Comment by R#2	Response
5. “We embed our analysis in the RCP-SSP framework: Specifically, the following three scenarios are analysed until 2050.” – is the AR6 the source – reference needed here rather than wait until the Methods.	The RCP-SSP framework has been developed by the climate change research community, so the framework has not been developed by the IPCC itself. We have added two references in the main text where the framework is mentioned the first time. Moss et al. (2010) describe the framework in general and the selection process of RCPs, as well as O’Neill et al. (2017) who describe the SSPs in more detail.
6. “relative to current sea levels, i.e., 1995-2014” – these are NOT current sea levels – they are “relative to 1995-2014 sea levels”.	We have changed accordingly.

Reviewer comments, third round review –

Reviewer #1 (Remarks to the Author):

The authors have thoroughly addressed my remaining concerns and I have no further suggestions.

Reviewer #2 (Remarks to the Author):

The authors have responded to all my comments to my satisfaction and I am happy for the paper to be published.

However, editorially, the language could still be improved.

Reviewer #1 (Remarks to the Author):

The authors have thoroughly addressed my remaining concerns and I have no further suggestions.

→ *Answer by authors: many thanks for reviewing our manuscript.*

Reviewer #2 (Remarks to the Author):

The authors have responded to all my comments to my satisfaction and I am happy for the paper to be published.

→ *Answer by authors: many thanks for reviewing our manuscript.*

However, editorially, the language could still be improved.

→ *Answer by authors: a native speaker has again checked the final manuscript for language and grammar.*